# VAT-Mart: Learning Visual Action Trajectory Proposals for Manipulating 3D Articulated Objects

**Ruihai Wu**[1*]   **Yan Zhao**[1*]   **Kaichun Mo**[3*]   **Zizheng Guo**[1]   **Yian Wang**[1]
**Tianhao Wu**[1]   **Qingnan Fan**[4]   **Xuelin Chen**[4]   **Leonidas Guibas**[3]   **Hao Dong**[1,2†]
[1]CFCS, CS Dept., PKU   [2]AIIT, PKU   [3]Stanford University   [4]Tencent AI Lab
{wuruihai,zhaoyan790,gzz,yianwang,hao.dong}@pku.edu.cn,
thwu@stu.pku.edu.cn,
{kaichun,guibas}@cs.stanford.edu,
{fqnchina,xuelin.chen.3d}@gmail.com

## Abstract

Perceiving and manipulating 3D articulated objects (*e.g.*, cabinets, doors) in human environments is an important yet challenging task for future home-assistant robots. The space of 3D articulated objects is exceptionally rich in their myriad semantic categories, diverse shape geometry, and complicated part functionality. Previous works mostly abstract kinematic structure with estimated joint parameters and part poses as the visual representations for manipulating 3D articulated objects. In this paper, we propose *object-centric actionable visual priors* as a novel perception-interaction handshaking point that the perception system outputs more actionable guidance than kinematic structure estimation, by predicting *dense geometry-aware*, *interaction-aware*, and *task-aware* visual action affordance and trajectory proposals. We design an *interaction-for-perception* framework VAT-Mart to learn such actionable visual representations by simultaneously training a curiosity-driven reinforcement learning policy exploring diverse interaction trajectories and a perception module summarizing and generalizing the explored knowledge for pointwise predictions among diverse shapes. Experiments prove the effectiveness of the proposed approach using the large-scale PartNet-Mobility dataset in SAPIEN environment and show promising generalization capabilities to novel test shapes, unseen object categories, and real-world data.

## 1 Introduction

We live in a 3D world composed of a plethora of 3D objects. To help humans perform everyday tasks, future home-assistant robots need to gain the capabilities of perceiving and manipulating a wide range of 3D objects in human environments. Articulated objects that contain functionally important and semantically interesting articulated parts (*e.g.*, cabinets with drawers and doors) especially require significantly more attention, as they are more often interacted with by humans and artificial intelligent agents. Having much higher degree-of-freedom (DoF) state spaces, articulated objects are, however, generally more difficult to understand and subsequently to interact with, compared to 3D rigid objects that have only 6-DoF for their global poses.

There has been a long line of research studying the perception and manipulation of 3D articulated objects in computer vision and robotics. On the perception side, researchers have developed various successful visual systems for estimating kinematic structures (Abbatematteo et al., 2019; Staszak et al., 2020), articulated part poses (Li et al., 2020; Jain et al., 2020; Liu et al., 2020), and joint parameters (Wang et al., 2019b; Yan et al., 2019). Then, with these estimated visual articulation models, robotic manipulation planners and controllers can be leveraged to produce action trajectories for robot executions (Klingbeil et al., 2010; Arduengo et al., 2019; Florence et al., 2019; Urakami et al., 2019; Mittal et al., 2021). While the commonly used two-stage solution underlying most of these systems reasonably breaks the whole system into two phases and thus allows bringing together

---

*Equal contribution
†Corresponding author; Project page: https://hyperplane-lab.github.io/vat-mart

Figure 1: Given an input 3D articulated object (a), we propose a novel perception-interaction handshaking point for robotic manipulation tasks - *object-centric actionable visual priors*, including per-point *visual action affordance predictions* (b) indicating *where* to interact, and *diverse trajectory proposals* (c) for selected contact points (marked with green dots) suggesting *how* to interact.

well-developed techniques from vision and robotics communities, the current handshaking point – *the standardized visual articulation models* (*i.e.* kinematic structure, articulated part poses, and joint parameters), may not be the best choice, since essential geometric and semantic features for robotic manipulation tasks, such as interaction hotspots (*e.g.* edges, holes, bars) and part functionality (*e.g.* handles, doors), are inadvertently abstracted away in these canonical representations.

We propose a new type of *actionable visual representations* (Do et al., 2018; Nagarajan et al., 2019; Mo et al., 2021) exploring a more *geometry-aware*, *interaction-aware*, and *task-aware* perception-interaction handshaking point for manipulating 3D articulated objects. Concretely, we train the perception system to predict action possibility and visual action trajectory proposals at every point over parts of 3D articulated objects (See Figure 1). In contrast to previous work that use standardized visual articulation models as visual representations, our framework VAT-MART predicts per-point dense action trajectories that are adaptive to the change of geometric context (*e.g.*, handles, door edges), interactions (*e.g.*, pushing, pulling), and tasks (*e.g.*, open a door for 30°, close up a drawer by 0.1-unit-length). Abstracting away from concrete external manipulation environments, such as robot arm configurations, robot base locations, and scene contexts, we aim for learning unified *object-centric visual priors* with *a dense* and *diverse* superset of visual proposals that can be potentially applied to different manipulation setups, avoiding learning separate manipulation representations under different circumstances.

The proposed *actionable visual priors*, as a "preparation for future tasks" (Ramakrishnan et al., 2021) or "visually-guided plans" (Wang et al., 2019a; Karamcheti et al., 2021), can provide informative guidance for downstream robotic planning and control. Sharing a similar spirit with Nagarajan et al. (2019); Mo et al. (2021), we formulate our visual action possibility predictions as per-point affordance maps, on which the downstream robotic planners may sample a position to interact according to the predicted likelihood of success. Then, for a chosen point for interaction, the discrete task planner may search for applicable interaction modes (*e.g.*, whether to attempt a grasp) within a much smaller space formed by the visual action trajectory distribution, instead of searching in the entire solution space. Next, considering the robot kinematic constraints and physic collisions, the continuous motion planner can further select an open-loop trajectory from the set of proposed visual action trajectory candidates as an initial value for optimization, and finally pass to the robot controller for execution. More recent reinforcement learning (RL) based planners and controllers can also benefit from our proposed solution spaces for more efficient exploration.

To obtain such desired *actionable visual priors*, we design an *interaction-for-perception* learning framework VAT-MART, as shown in Figure 2. By conducting trial-and-error manipulation with a set of diverse 3D articulated objects, we train an RL policy to learn successful interaction trajectories for accomplishing various manipulation tasks (*e.g.*, open a door for 30°, close up a drawer by 0.1-unit-length). In the meantime, the perception networks are simultaneously trained to summarize the RL discoveries and generalize the knowledge across points over the same shape and among various shapes. For discovering diverse trajectories, we leverage curiosity feedback (Pathak et al., 2017) for enabling the learning of perception networks to reversely affect the learning of RL policy.

We conduct experiments using SAPIEN (Xiang et al., 2020) over the large-scale PartNet-Mobility (Chang et al., 2015; Mo et al., 2019) dataset of 3D articulated objects. We use 562 shapes in 7 object categories to perform our experiments and show that our VAT-MART framework can successfully learn the desired *actionable visual priors*. We also observe reasonably good gen-

eralization capabilities over unseen shapes, novel object categories, and real-world data, thanks to large-scale training over diverse textureless geometry.

In summary, we make the following contributions in this work:

- We formulate a novel kind of *actionable visual priors* making one more step towards bridging the perception-interaction gap for manipulating 3D articulated objects;
- We propose an *interaction-for-perception* framework VAT-MART to learn such priors with novel designs on the joint learning between exploratory RL and perception networks;
- Experiments conducted over the PartNet-Mobility dataset in SAPIEN demonstrate that our system works at a large scale and learns representations that generalize over unseen test shapes, across object categories, and even real-world data.

## 2 RELATED WORK

**Perceiving and Manipulating 3D Articulated Objects** has been a long-lasting research topic in computer vision and robotics. A vast literature (Yan & Pollefeys, 2006; Katz et al., 2008; Sturm et al., 2009; 2011; Huang et al., 2012; Katz et al., 2013; Martin & Brock, 2014; Höfer et al., 2014; Katz et al., 2014; Schmidt et al., 2014; Hausman et al., 2015; Martín-Martín et al., 2016; Tzionas & Gall, 2016; Paolillo et al., 2017; Martın-Martın & Brock, 2017; Paolillo et al., 2018; Martín-Martín & Brock, 2019; Desingh et al., 2019; Nunes & Demiris, 2019) has demonstrated successful systems, powered by visual feature trackers, motion segmentation predictors, and probabilistic estimators, for obtaining accurate link poses, joint parameters, kinematic structures, and even system dynamics of 3D articulated objects. Previous works (Peterson et al., 2000; Jain & Kemp, 2009; Chitta et al., 2010; Burget et al., 2013) have also explored various robotic planning and control methods for manipulating 3D articulated objects. More recent works further leveraged learning techniques for better predicting articulated part configurations, parameters, and states (Wang et al., 2019b; Yan et al., 2019; Jain et al., 2020; Zeng et al., 2020; Li et al., 2020; Liu et al., 2020; Mu et al., 2021), estimating kinematic structures (Abbatematteo et al., 2019; Staszak et al., 2020), as well as manipulating 3D articulated objects with the learned visual knowledge (Klingbeil et al., 2010; Arduengo et al., 2019; Florence et al., 2019; Urakami et al., 2019; Mittal et al., 2021). While most of these works represented visual data with link poses, joint parameters, and kinematic structures, such standardized abstractions may be insufficient if fine-grained part geometry, such as drawer handles and faucet switches that exhibit rich geometric diversity among different shapes, matters for downstream robotic tasks.

**Learning Actionable Visual Representations** aims for learning visual representations that are strongly aware of downstream robotic manipulation tasks and directly indicative of action probabilities for robotic executions, in contrast to predicting standardized visual semantics, such as category labels (Russakovsky et al., 2015; Wu et al., 2015b), segmentation masks (Lin et al., 2014; Mo et al., 2019), and object poses (Hinterstoisser et al., 2011; Xiang et al., 2016), which are usually defined independently from any specific robotic manipulation task. Grasping (Montesano & Lopes, 2009; Lenz et al., 2015; Mahler et al., 2017; Fang et al., 2020; Mandikal & Grauman, 2021; Corona et al., 2020; Kokic et al., 2020; Yang et al., 2020; Jiang et al., 2021) or manipulation affordance (Kjellström et al., 2011; Do et al., 2018; Fang et al., 2018; Goff et al., 2019; Nagarajan et al., 2019; Nagarajan & Grauman, 2020; Nagarajan et al., 2020; Xu et al., 2021; Mo et al., 2021) is one major kind of actionable visual representations, while many other types have been also explored recently (*e.g.*, spatial maps (Wu et al., 2020; 2021), keypoints (Wang et al., 2020; Qin et al., 2020), contact points (You et al., 2021), etc). Following the recent work Where2Act (Mo et al., 2021), we employ dense affordance maps as the actionable visual representations to suggest action possibility at every point on 3D articulated objects. Extending beyond Where2Act which considers task-less short-term manipulation, we further augment the per-point action predictions with task-aware distributions of trajectory proposals, providing more actionable information for downstream executions.

**Learning Perception from Interaction** augments the tremendously successful learning paradigm using offline curated datasets (Russakovsky et al., 2015; Lin et al., 2014; Chang et al., 2015; Mo et al., 2019) by allowing learning agents to collect online active data samples, which are more task-aware and learning-efficient, during navigation (Anderson et al., 2018; Ramakrishnan et al., 2021), recognition (Wilkes & Tsotsos, 1992; Yang et al., 2019; Jayaraman & Grauman, 2018), segmentation (Pathak et al., 2018; Gadre et al., 2021), and manipulation (Pinto et al., 2016; Bohg et al., 2017). Many works have also demonstrated the usefulness of simulated interactions for learning perception (Wu et al., 2015a; Mahler et al., 2017; Xu et al., 2019; Ramakrishnan et al., 2021;

Nagarajan & Grauman, 2020; Lohmann et al., 2020; Mo et al., 2021) and promising generalizability to the real world (James et al., 2019; Chebotar et al., 2019; Hundt et al., 2019; Liang et al., 2020; Kadian et al., 2020; Anderson et al., 2020; Rao et al., 2020). Our method follows the route of learning perception from interaction via using the action trajectories discovered by an RL interaction policy to supervise a jointly trained perception system, which reversely produces curiosity feedback (Pathak et al., 2017) to encourage the RL policy to explore diverse action proposals.

## 3 ACTIONABLE VISUAL PRIORS: *Action Affordance and Trajectory Proposals*

We propose novel *actionable visual representations* for manipulating 3D articulated objects (see Fig. 1). For each articulated object, we learn *object-centric actionable visual priors*, which are comprised of: 1) an *actionability map* over articulated parts indicating *where* to interact; 2) per-point distributions of *visual action trajectory proposals* suggesting *how* to interact; and 3) estimated success likelihood scores rating the *outcomes* of the interaction. All predictions are *interaction-conditioned* (*e.g.*, pushing, pulling) and *task-aware* (*e.g.*, open a door for $30°$, close a drawer by 0.1-unit-length).

Concretely, given a 3D articulated object $O$ with its articulated parts $\mathscr{P} = \{P_1, P_2, \cdots\}$, an interaction type $T$, and a manipulation task $\theta$, we train a perception system that makes *dense* predictions at each point $p$ over each articulated part $p \in \cup \mathscr{P}$: 1) an actionability score $a_{p|O,T,\theta} \in [0,1]$ indicating how likely there exists an action trajectory of interaction type $T$ at point $p$ that can successfully accomplish the task $\theta$; 2) a distribution of *visual action trajectories* $\mathbb{P}_p(\cdot|O,T,\theta)$, from which we can sample *diverse* action trajectories $\tau_{p|O,T,\theta} \sim \mathbb{P}_p(\cdot|O,T,\theta)$ of interaction type $T$ to accomplish the task $\theta$ at point $p$; and 3) a per-trajectory *success likelihood score* $r_{\tau|O,p,T,\theta} \in [0,1]$.

**Inputs.** We represent the input 3D articulated object $O$ as a partial point cloud $S_O$. We consider two typical interaction types: *pushing* and *pulling*. A *pushing* trajectory maintains a closed gripper and has 6-DoF motion performing the pushing, whereas a *pulling* trajectory first performs a grasping operation at the point of interaction by closing an initially opened gripper and then has the same 6-DoF motion during the pulling. For articulated objects we use in this work, we only consider 1-DoF part articulation and thus restrict the task specification $\theta \in \mathbb{R}$. For example, a cabinet drawer has a 1-DoF prismatic translation-joint and a refrigerator door is modeled by a 1-DoF revolute hinge-joint. We use the absolute angular degrees in radian (*i.e.* $\theta \in [-\pi, \pi]$) for revolute joints and use the units of length (*i.e.* $\theta \in [-1, 1]$) relative to the global shape scale for prismatic joints.

**Outputs.** Both the actionability score $a_{p|O,T,\theta}$ and per-trajectory success likelihood score $r_{\tau|O,p,T,\theta}$ are scalars within $[0,1]$, where larger values indicate higher likelihood. One can use a threshold of 0.5 to obtain binary decisions if needed. Every action trajectory $\tau_{p|O,T,\theta}$ is a sequence of 6-DoF end-effector waypoints $(wp_0, wp_1, \cdots, wp_k)$, with variable trajectory length ($k \leq 5$). In our implementation, we adopt a residual representation $(wp_0, wp_1 - wp_0, \cdots, wp_k - wp_{k-1})$ for the action trajectory, as it empirically yields better performance. Each 6-DoF waypoint is comprised of a 3-DoF robot hand center $x$ and 3-DoF orientation $R$. We use the 6D-rotation representation (Zhou et al., 2019) for the orientation of $wp_0$ and predict 3-DoF euler angles for subsequent orientation changes.

## 4 VAT-MART: AN *Interaction-for-perception* LEARNING FRAMEWORK

The VAT-MART system (Fig. 2) consists of two parts: an RL policy exploring diverse action trajectories and a perception system learning the proposed *actionable visual priors*. While the RL policy collects interaction trajectories for supervising the perception networks, the perception system provides curiosity feedback (Pathak et al., 2017) for encouraging the RL policy to further explore diverse solutions. In our implementation, we first pretrain the RL policy, then train the perception network with RL-collected data, and finally finetune the two parts jointly with curiosity-feedback enabled. We describe key system designs below and will release code for our implementation.

### 4.1 THE RL POLICY FOR INTERACTIVE TRAJECTORY EXPLORATION

For every interaction type $T$ and part articulation type, we train a single conditional RL policy using TD3 (Fujimoto et al., 2018) to collect trajectories $\tau$ that can accomplish the interaction of varying task specifications $\theta$ across all shapes $O$ and contact points $p$. In the RL training, since the RL policy is trained in the simulation for only collecting training data to supervise the perception networks, we can have access to the ground-truth state information of the simulation environment, such as the part

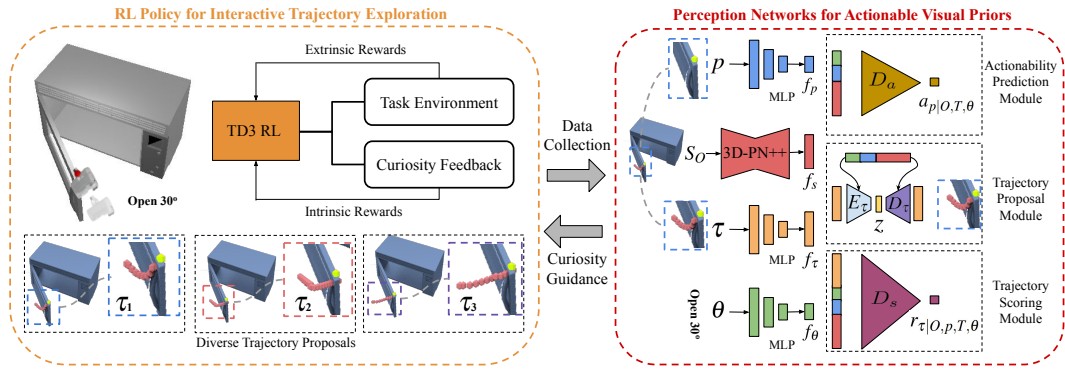

Figure 2: Our proposed VAT-MART framework is composed of an RL policy (left) exploring interaction trajectories and a perception system (right) learning the desired actionable visual priors. We build bidirectional supervisory channels between the two parts: 1) the RL policy collects data to supervise the perception system, and 2) the perception system produces curiosity feedbacks encouraging the RL networks to explore diverse solutions.

poses, joint axis, and gripper poses. At the test time, we discard the RL network and only use the learned perception networks to predict the proposed *actionable visual priors*.

Fig. 2 (left) illustrates the RL training scheme and example explored diverse trajectory proposals for the task of pulling open the microwave door for 30 degrees. Below, we describe the RL specifications.

**Task Initialization.** For the shape to interact with, we first randomly sample a shape category with equal probability, alleviating the potential category imbalance issue, and then uniformly sample a shape from the selected category for training. For the task specification $\theta$, we randomly sample within $[10°, 70°]$ for revolute parts and $[0.1, 0.7]$ for prismatic parts. We also randomly sample a starting part pose $\theta_0$ with the guarantee that the task $\theta$ can be accomplished. For the gripper, we initialize it with fingertip 0.02-unit-length away from the contact point $p$ and pointing within a cone of 30 degrees along the negative direction of the surface normal at $p$.

**State Space.** The RL state includes the 1-DoF part pose change $\Delta\theta_i = \theta_i - \theta_0$ at the current timestep, the target task $\theta$, the difference $\theta - \Delta\theta_i$, the gripper pose $wp_0 = (x_0, R_0) \in SE(3)$ at the first timestep of interaction, the current gripper pose $wp_i = (x_i, R_i) \in SE(3)$, the local positions for the gripper fingers $x_f \in \mathbb{R}^2$, the current contact point location $p_i \in \mathbb{R}^3$, a normalized direction of the articulated part joint axis $n_j \in \mathbb{R}^3$, the articulated part joint location $x_j \in \mathbb{R}^3$ (defined as the closest point on the joint axis to the contact point $p$), the closest distance from the contact point to the joint axis $d_{cj} \in \mathbb{R}$, and a directional vector $n_{cj} \in \mathbb{R}^3$ from the joint location to the contact point. We concatenate all the information together as a 33-dimensional state vector for feeding to the RL networks.

**Action Space.** At each timestep, we predict a residual gripper pose $wp_i - wp_{i-1} \in SE(3)$ to determine the next-step waypoint $wp_i$ as the action output of the RL networks. We estimate a center offset $x_i - x_{i-1} \in \mathbb{R}^3$ and an euler angle difference $R_i - R_{i-1} \in SO(3)$.

**Reward Design.** There are two kinds of rewards: extrinsic task rewards and intrinsic curiosity feedbacks. For the extrinsic task rewards, we use: 1) a final-step success reward of 500 for a task completion when the current part pose reaches the target within 15% relative tolerance range, 2) a step-wise guidance reward of $300(|\theta - \Delta\theta_{i-1}| - |\theta - \Delta\theta_i|)$ encouraging the current part pose to get closer to the target than previous part pose, and 3) a distance penalty of $300 \cdot \mathbb{1}[d_{gc} > 0.1] + 150 d_{gc}$ to discourage the gripper from flying away from the intended contact point $p$, where $d_{gc}$ denotes the $l_2$ distance from the contact point $p$ to the current fingertip position and $\mathbb{1}[q]$ is a zero or one function indicating the boolean value of the predicate $q$. We will describe the curiosity rewards in Sec. 4.3.

**Stopping Criterion.** We stop an interaction trial until the task's success or after five maximal waypoint steps.

**Implementation and Training.** We implement the TD3 networks using MLPs and use a replay buffer of size 2048. To improve the positive data rates for efficient learning, we leverage Hindsight Experience Replay (Andrychowicz et al., 2017): an interaction trial may fail to accomplish the desired task $\theta$, but it finally achieves the task of $\theta_k - \theta_0$. See Sec. A in appendix for more details.

## 4.2 The Perception Networks for Actionable Visual Priors

The perception system learns from the interaction trajectories collected by the RL exploration policy and predicts the desired *per-pixel actionable visual priors*. Besides several information encoding modules, there are three decoding heads: 1) an actionability prediction module that outputs the actionability score $a_{p|O,\theta} \in [0,1]$, 2) a trajectory proposal module that models per-point distribution of diverse visual action trajectories $\tau_{p|O,T,\theta} = (wp_0, wp_1 - wp_0, \cdots, wp_k - wp_{k-1})$, and 3) a trajectory scoring module that rates the per-trajectory success likelihood $r_{\tau|O,p,T,\theta} \in [0,1]$. Fig. 2 (right) presents an overview of the system. Below we describe detailed designs and training strategies.

**Input Encoders.** The perception networks require four input entities: a partial object point cloud $S_O$, a contact point $p$, a trajectory $\tau$, and a task $\theta$. For the point cloud input $S_O$, we use a segmentation-version PointNet++ (Qi et al., 2017) to extract a per-point feature $f_s \in \mathbb{R}^{128}$. We employ three MLP networks that respectively encode the inputs $p$, $\tau$, and $\theta$ into $f_p \in \mathbb{R}^{32}$, $f_\tau \in \mathbb{R}^{128}$, and $f_\theta \in \mathbb{R}^{32}$. We serialize each trajectory as a 30-dimensional vector after flattening all waypoint information. We augment the trajectories that are shorter than five waypoint steps simply by zero paddings.

**Actionability Prediction Module.** The actionability prediction network $D_a$, similar to Where2Act (Mo et al., 2021), is implemented as a simple MLP that consumes a feature concatenation of $f_p$, $f_s$, and $f_\theta$, and predicts a per-point actionability score $a_{p|O,T,\theta} \in [0,1]$. Aggregating over all contact points $p$, one can obtain an actionability map $A_{O,T,\theta}$ over the input partial scan $S_O$, from which one can sample an interaction point at test time according to a normalized actionability distribution.

**Trajectory Proposal Module.** The trajectory proposal module is implemented as a conditional variational autoencoder (cVAE, Sohn et al. (2015)), composed of a trajectory encoder $E_\tau$ that maps the input trajectory $\tau$ into a Gaussian noise $z$ and a trajectory decoder $D_\tau$ that reconstructs the trajectory input from the noise vector. Both networks take additional input features of $f_p$, $f_s$, and $f_\theta$ as conditions. We use MLPs to realize the two networks. We regularize the resultant noise vectors to get closer to a uniform Gaussian distribution so that one can sample diverse trajectory proposals by feeding random Gaussian noises to the decoder $D_\tau$ with the conditional features as inputs.

**Trajectory Scoring Module.** The trajectory scoring module $D_s$, implemented as another MLP, takes as inputs features of $f_p$, $f_s$, and $f_\theta$, as well as the trajectory feature $f_\tau$, and predicts the success likelihood $r_{\tau|O,p,T,\theta} \in [0,1]$. One can use a success threshold of 0.5 to obtain a binary decision.

**Data Collection for Training.** We collect interaction data from the RL exploration to supervise the training of the perception system. We randomly pick shapes, tasks, and starting part poses similar to the RL task initialization. For positive data, we sample 5000 successful interaction trajectories outputted by the RL. We sample the same amount of negative data, which are produced by offsetting the desired task $\theta_0$ of a successful trajectory by a random value with $[0.1\theta_0, 45°]$ for revolute parts and $[0.1\theta_0, 0.45]$ for prismatic parts. For the pulling experiments, we also consider another type of negative data that the first grasping attempt fails.

**Implementation and Training.** We train the trajectory scoring and the trajectory proposal modules before the actionability prediction module. We use the standard binary cross-entropy loss to train the trajectory scoring module $D_s$. To train the trajectory proposal cVAE of $E_\tau$ and $D_\tau$, besides the KL divergence loss for regularizing Gaussian bottleneck noises, we use an $L_1$ loss to regress the trajectory waypoint positions and a 6D-rotation loss (Zhou et al., 2019) for training the waypoints orientations. For training the actionability prediction module $D_a$, we sample 100 random trajectories proposed by $D_\tau$, estimate their success likelihood scores using $D_s$, and regress the prediction to the mean score of the top-5 rated trajectories with a $L_1$ loss. See Sec. B in appendix for more details.

## 4.3 Curiosity-driven Exploration

We build a bidirectional supervisory mechanism between the RL policy and the perception system. While the RL policy collects data to supervise the perception networks, we also add a curiosity-feedback (Pathak et al., 2017) from the perception networks to inversely affect the RL policy learning for exploring more diverse and novel interaction trajectories, which will eventually diversify the per-point trajectory distributions produced by the trajectory proposal decoder $D_\tau$. The intuitive idea is to encourage the RL network to explore novel trajectories that the perception system currently gives low success scores. In our implementation, during the joint training of the RL and perception networks, we generate an additional intrinsic curiosity reward of $-500 r_{\tau|O,p,T,\theta}$ for a trajectory $\tau$

Table 1: We report quantitative evaluations of the learned *actionable visual priors*. For each metric, we report numbers over test shapes from the training categories (before slash) and shapes from the test categories (after slash). Higher numbers indicate better results.

| | | Accuracy (%) | Precision (%) | Recall (%) | F-score (%) | Coverage (%) |
|---|---|---|---|---|---|---|
| door | pushing | 82.24 / 72.44 | 81.28 / 72.83 | 85.22 / 73.86 | 82.76 / 72.54 | 82.00 / 70.54 |
| | pulling | 74.01 / 71.31 | 70.52 / 70.26 | 84.09 / 75.85 | 76.06 / 72.01 | 58.68 / 48.29 |
| drawer | pushing | 79.69 / 71.59 | 74.65 / 71.80 | 91.19 / 70.45 | 81.65 / 70.52 | 74.15 / 68.08 |
| | pulling | 78.41 / 71.88 | 74.54 / 72.29 | 87.50 / 72.44 | 80.23 / 71.71 | 81.15 / 64.31 |

Figure 3: We show qualitative results of the actionability prediction and trajectory proposal modules. In each result block, from left to right, we present the input shape with the task, the predicted actionability heatmap, and three example trajectory proposals at a selected contact point.

proposed by the RL policy and use the novel interaction data to continue supervising the perception system. We make sure to have training trajectory proposals generated by the RL network at different epochs in the buffer to avoid mode chasing of training the generative model $D_\tau$.

## 5 EXPERIMENTS

We perform our experiments using the SAPIEN simulator (Xiang et al., 2020) and PartNet-Mobility dataset (Chang et al., 2015; Mo et al., 2019). We evaluate the prediction quality of the learned visual priors and compare them to several baselines for downstream manipulation tasks. Qualitative results over novel shapes and real-world data show promising generalization capability of our approach.

**Data and Settings.** In total, we use 562 shapes from 7 categories in the PartNet-Mobility dataset. We conduct experiments over two commonly seen part articulation types: doors and drawers. For each experiment, we randomly split the applicable object categories into training and test categories. We further split the data from the training categories into training and test shapes. We train all methods over training shapes from the training categories and report performance over test shapes from the training categories and shapes from the test categories to test the generalization capabilities over novel shapes and unseen categories. We use the Panda flying gripper as the robot actuator and employ a velocity-based PID-controller to realize the actuation torques between consecutive trajectory waypoints. We use an RGB-D camera of resolution 448 × 448 and randomly sample viewpoints in front of the objects. See Sec. D E in appendix for more data statistics and implementation details.

### 5.1 ACTIONABLE VISUAL PRIORS

We present qualitative and quantitative evaluations of our learned actionable visual priors.

**Metrics.** For quantitatively evaluating the trajectory scoring module, we use standard metrics that are commonly used for evaluating binary classification problems: accuracy, precision, recall, and F-score. Since there is no ground-truth annotation of successful and failed interaction trajectories, we run our learned RL policy over test shapes and collect test interaction trajectories. In our implementation, we gather 350 positive and 350 negative trajectories for each experiment. To evaluate if the learned trajectory proposal module can propose diverse trajectories to cover the collected ground-truth ones,

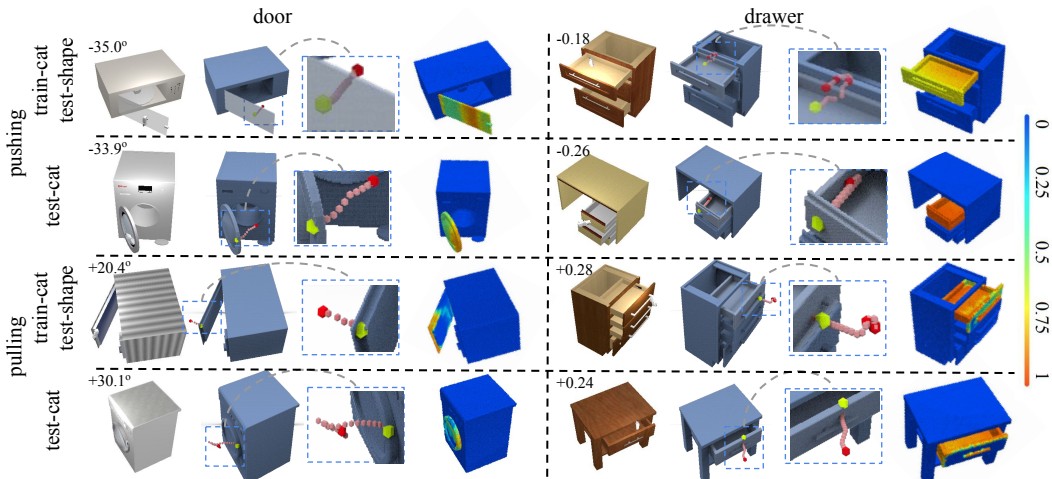

Figure 4: We present qualitative analysis of the learned trajectory scoring module. In each result block, from left to right, we show the input shape with the task, the input trajectory with its close-up view, and our network predictions of success likelihood applying the trajectory over all the points.

we compute a coverage score measuring the percentage of ground-truth trajectories that is similar enough to the closest predicted trajectory. See Sec. G in appendix for detailed metric definitions.

**Results and Analysis.** Table 1 presents quantitative results which demonstrate that we successfully learn the desired representations and that our learned model also generalizes to shapes from totally unseen object categories. Since we are the first to propose such actionable visual priors, there is no baseline to compare against. Fig. 3 presents qualitative results of the actionability prediction and trajectory proposal modules, from which we can observe that, to accomplish the desired tasks: 1) the actionability heatmap highlights *where* to interact (*e.g.*, to pull open a drawer, the gripper either grasp and pull open the handle from outside, or push outwards from inside), and 2) the trajectory proposals suggest diverse solutions for *how* to interact (*e.g.*, to push closed the door, various trajectories may succeed). In Fig. 4, we additionally illustrate the trajectory scoring predictions using heatmap visualization, where we observe interesting learned patterns indicating which points to interact with when executing the input trajectories. Fig. 5 (left) further visualizes the trajectory scoring predictions over the same input shape and observe different visual patterns given different trajectories and tasks.

## 5.2 DOWNSTREAM MANIPULATION

We can easily use our learned actionable visual priors to accomplish the downstream manipulation of 3D articulated objects. To this end, we first sample a contact point according to the estimated action-ability heatmap and then execute the top-rated trajectory among 100 random trajectory proposals.

**Baselines.** We compare to three baselines: 1) a naive TD3 RL that takes the shape point cloud together with the desired task as input and directly outputs trajectory waypoints for accomplishing the task; 2) a heuristic approach, in which we hand-engineer a set of rules for different tasks (*e.g.*, to pull open a drawer, we grasp the handle and pull straight backward). Note that we use ground-truth handle masks and joint parameters for the heuristic baseline; 3) a multi-step Where2Act (Mo et al., 2021) baseline that concatenates multiple (up to 15 steps) Where2Act-generated short-term pushing or pulling interactions that gradually accumulates the part pose changes for accomplishing the task. While the Where2Act work originally considers task-less short-term interactions, different from our long-term task-driven setting, we adapt it as a baseline by assuming an oracle part pose tracker is given. See Sec. H in appendix for more detailed descriptions of the baseline methods.

**Metrics.** We run interaction trials in simulation and report success rates for quantitative evaluation.

**Results and Analysis.** Table 2 presents the quantitative comparisons. Our method outperforms baselines on most comparisons. See below for detailed result analysis and more results in appendix.

For **the naive RL baseline**, it largely fails since we find it extremely difficult to train from scratch an end-to-end RL over highly diverse shapes for all tasks, which in fact implies the necessity of certain intermediate visual abstractions.

Table 2: We report task success rates (within 15% tolerance to tasks) of manipulating articulated objects comparing our method against three baseline methods.

|  | pushing door | pulling door | pushing drawer | pulling drawer |
|---|---|---|---|---|
| RL-baseline | 1.43 / 3.72 | 0.0 / 0.0 | 0.43 / 0.43 | 0.0 / 0.0 |
| Heuristic | 25.74 / 25.23 | 8.82 / 3.73 | 40.25 / **57.99** | 25.02 / 22.85 |
| Where2Act | 32.67 / 32.27 | 6.02 / 3.51 | 29.79 / 23.67 | 8.31 / 9.71 |
| Ours | **55.14 / 36.49** | **12.04 / 14.33** | **56.02** / 38.28 | **43.53 / 31.14** |

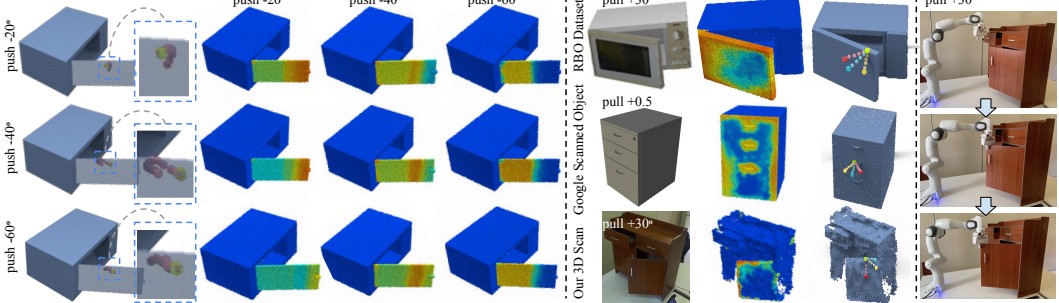

Figure 5: Left: qualitative analysis of the trajectory scoring prediction (each column shares the same task; every row uses the same trajectory); Middle: promising results testing on real-world data (from left to right: input, affordance prediction, trajectory proposals); Right: real-robot experiment.

For **the heuristic baseline**, we find that we win all comparisons except one task of drawer pushing. Knowing the ground-truth front board and prismatic joint axis, it is quite easy for the rule-based method to push a drawer along a straight-line trajectory, while our method does not take the ground-truth information and achieves worse results since we have to predict such information. However, our method achieves better performance for the other three experiments since we find that the heuristic method fails (see Fig. H.7) severely when 1) there are no handle parts for pulling, 2) some grasps over intricate handle geometry may slip and fail, or 3) the motion dynamics (*e.g.*, inertia) for non-prehensile manipulation (pushing) affects the accurate task accomplishment (*e.g.*, in case of an overshoot).

For **the Where2Act baseline**, it consistently performs worse than ours since it does not explicitly take the desired task as input and thus is more vulnerable of overshooting. Besides, the Where2Act baseline has the following issues: 1) it assumes an accurate part pose tracker, 2) it has to execute multiple segmented interactions and thus is empirically found 30 times slower, and 3) the gripper needs to be driven to multiple independent positions across different trajectory segments and thus it is very time-consuming and fragile if cases of motion planning failures.

### 5.3 REAL-WORLD EXPERIMENTS

Fig. 5 (middle) presents qualitative results directly testing our model on real-world data: a microwave from the RBO dataset (Martín-Martín et al., 2019), one cabinet from Google Scanned Object (Open-Robotics), and one real-world 3D cabinet scan we capture using a ZED MINI RGB-D camera. We observe that our model trained on synthetic textureless data can generalize to real-world depth scans to some degree. We also show real-robot experiment in Fig. 5 (right) and supplementary video.

## 6 CONCLUSION

In this paper, we propose a novel perception-interaction handshaking point –*object-centric actionable visual priors*– for manipulating 3D articulated objects, which contains *dense* action affordance predictions and *diverse* visual trajectory proposals. We formulate a novel *interaction-for-perception* framework VAT-MART to learn such representations. Experiments conducted on the large-scale PartNet-Mobility dataset and real-world data have proved the effectiveness of our approach.

**Limitations and Future Works.** First, our work is only a first attempt at learning such representations and future works can further improve the performance. Besides, the current open-loop trajectory prediction is based on a single-frame input. One may obtain better results considering multiple frames during an interaction. Lastly, future works may study more interaction and articulation types.

**Acknowledgements.** National Natural Science Foundation of China —Youth Science Fund (No.62006006): Learning Visual Prediction of Interactive Physical Scenes using Unlabelled Videos. Leonidas and Kaichun are supported by a grant from the Toyota Research Institute University 2.0 program[1], NSF grant IIS-1763268, and a Vannevar Bush faculty fellowship. We thank Yourong Zhang for setting up ROS environment and helping in real robot experiments, Jiaqi Ke for designing and doing heuristic experiments.

**Ethics Statement.** Our project helps build home-assistant robots to aid people with disabilities, or senior. Our proposed visual affordance and action trajectory predictions can be visualized and analyzed before use, which makes the AI system more explainable and thus less vulnerable to potential dangers. Training our system requires training data, which may introduce data bias, but this is a general concern for learning-based methods. We do not see any particular major harm or issue our work may raise up.

**Reproducibility Statement.** We will release our code, data, and pre-trained model weights publicly to the community upon paper acceptance.

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

## APPENDIX

## A  MORE DETAILS FOR THE RL POLICY (SEC. 4.1)

### A.1  NETWORK ARCHITECTURE DETAILS

We leverage TD3 (Fujimoto et al., 2018) to train the RL policy. It consists of a policy network and a Q-value network, both implemented as a Multi-layer Perception (MLP). The policy network receives the state as input ($\mathbb{R}^{33}$), and predicts the residual gripper pose as action ($\mathbb{R}^6$). Its network architecture is implemented as a 4-layer MLP ($33 \rightarrow 512 \rightarrow 512 \rightarrow 512 \rightarrow 512$), followed by two separate fully connected layers ($512 \rightarrow 6$) that estimate the mean and variance of action probabilities respectively. The Q-value network receives both the state and action as input ($\mathbb{R}^{39}$), and predicts a single Q value ($\mathbb{R}^1$). Its network architecture is implemented as a 4-layer MLP ($39 \rightarrow 512 \rightarrow 512 \rightarrow 512 \rightarrow 1$).

### A.2  MORE IMPLEMENTATION AND TRAINING DETAILS

We use the following hyper-parameters to train the RL policy: 2048 (buffer size); 512 (batch size); 0.0001 (learning rate of the Adam optimizer (Kingma & Ba, 2015) for both the policy and Q-value network); 0.1 (initial exploration range, decayed by 0.5 every 500 epochs.

## B  MORE DETAILS FOR THE PERCEPTION NETWORKS (SEC. 4.2)

### B.1  NETWORK ARCHITECTURE DETAILS

The perception networks consist of four input encoders that extract high-dimensional features from the input partial point cloud $S_O$, contact point $p$, trajectory $\tau$ and task $\theta$, and three parallel output modules that estimate the actionability $a_{p|O,T,\theta}$, trajectory proposal $\tau_{p|O,T,\theta}$ and scores $r_{\tau|O,p,T,\theta}$ from the extracted high-dimensional features of input encoders.

Regarding the four input encoders

- The input point cloud $S_O$ with point-wise coordinate ($\mathbb{R}^3$) is fed through the PointNet++ (Qi et al., 2017) with the segmentation head and single-scale grouping (SSG) to extract the point-wise feature $f_s \in \mathbb{R}^{128}$.

- The input contact point $p \in \mathbb{R}^3$ is fed through a fully connected layer ($3 \rightarrow 32$) to extract a high-dimensional contact point feature $f_p \in \mathbb{R}^{32}$.

- The input trajectory $\tau \in \mathbb{R}^{30}$ is composed of five waypoints, each of which contains 3D rotation and 3D translation information. The input trajectory is further flattened to be fed through a 3-layer MLP ($30 \rightarrow 128 \rightarrow 128 \rightarrow 128$) to extract a high-dimensional trajectory feature $f_\tau \in \mathbb{R}^{128}$.

- The input task $\theta \in \mathbb{R}^1$ is fed through a fully connected layer ($1 \rightarrow 32$) to extract a high-dimensional task feature $f_\theta \in \mathbb{R}^{32}$.

Regarding the three output modules

- The actionability prediction module $D_a$ concatenates the extracted point cloud $f_s$, contact point $f_p$ and task $f_\theta$ features to form a high-dimensional vector ($\mathbb{R}^{192}$), which is fed through a 5-layer MLP ($192 \rightarrow 128 \rightarrow 128 \rightarrow 128 \rightarrow 128 \rightarrow 1$) to predict a per-point actionability score $a_{p|O,T,\theta}$.
- The trajectory proposal module is implemented as a conditional variational auto-encoder (cVAE), which learns to generate diverse trajectories from the sampled Gaussian value and given conditions (point cloud $f_s$, contact point $f_p$ and task $f_\theta$ features). The encoder $E_\tau$ is implemented as a 3-layer MLP ($320 \rightarrow 128 \rightarrow 128 \rightarrow 256$) that takes the concatenation of the trajectory feature $f_\tau$ and given conditions as input and estimates a Gaussian distribution ($\mu \in \mathbb{R}^{128}, \sigma \in \mathbb{R}^{128}$). The decoder $D_\tau$ is also implemented as a 3-layer MLP ($320 \rightarrow 512 \rightarrow 256 \rightarrow 30$) that takes the concatenation of the sampled Gaussian value $z \in \mathbb{R}^{128}$ and given conditions as input to recover the trajectory $\tau$.
- The trajectory scoring module $D_s$ concatenates the extracted point cloud $f_s$, contact point $f_p$, trajectory $f_\tau$ and task $f_\theta$ features to form a high-dimensional vector ($\mathbb{R}^{320}$), which is fed through a 2-layer MLP ($320 \rightarrow 128 \rightarrow 1$) to predict the success likelihood $r_{\tau|O,p,T,\theta}$.

## B.2 More Implementation and Training Details

The interaction data provided by the RL policy provides the ground truth label to train the perception networks. We start by training the trajectory scoring module, followed by joint training of all the three modules.

To train the trajectory scoring module $D_s$, we firstly collect 5000 successful and 5000 unsuccessful interaction trajectories produced by the RL policy, then supervise the estimated success likelihood of the trajectory with binary cross entropy loss.

To train the trajectory proposal module, we supervise the output rotation and translation with the 6D-rotation loss (Zhou et al., 2019) and $L_1$ loss respectively from the 5000 successful collected interaction trajectories. We follow the common cVAE training process to add a KL divergence loss on the estimated Gaussian distribution for regularization.

To train the actionability prediction module $D_a$, we estimate the scores from 100 sampled trajectories with both the trajectory proposal and scoring modules, then calculate the mean value of top-5 rated scores as the ground truth label to supervise the actionability predictions. We adopt $L_1$ loss for supervision.

We use the following hyper-parameters to train the perception networks: 32 (batch size); 0.001 (learning rate of the Adam optimizer (Kingma & Ba, 2015)) for all three modules.

## C More Details for the Curiosity-driven Exploration (Sec. 4.3)

The curiosity-driven exploration alternates between training the RL policy and perception networks at odd and even epochs respectively. It aims for exploring more diverse trajectories with the RL policy that the perception networks are not confident with and are further used to diversify the trajectory proposals generated by the perception networks.

When training the RL policy, besides the extrinsic task rewards, it also awards the policy with the intrinsic curiosity reward. It is computed as $-500 r_{\tau|O,p,T,\theta}$, which is the weighted negative success likelihood estimated by the trajectory scoring module. It penalizes the learned trajectories that the trajectory scoring module is confident with and encourage more diverse trajectories generated by the RL policy.

When training the perception networks, we collect the equal number of successful and unsuccessful trajectories produced by the RL policy. To encourage the perception networks to learn more diverse trajectories, we sample the successful trajectories fifty-fifty with both high (¿0.5) and low (¡0.5) success likelihood estimated by the trajectory scoring module. Then we use these data to train the perception networks.

| Train-Cats | Door / All | Cabinet | Door | Mirco | Fridge | Drawer / All | Cabinet |
|---|---|---|---|---|---|---|---|
| Train-Data | 328 | 270 | 17 | 9 | 32 | 270 | 270 |
| Test-Data | 94 | 75 | 5 | 3 | 11 | 75 | 75 |
| Test-Cats | | Safe | Table | Washing | | | Table |
| Test-Data | 140 | 29 | 95 | 16 | | 95 | 95 |

Table D.3: We summarize the shape counts in our dataset. Here, *Micro* and *Washing* are short for microwave and washing machine.

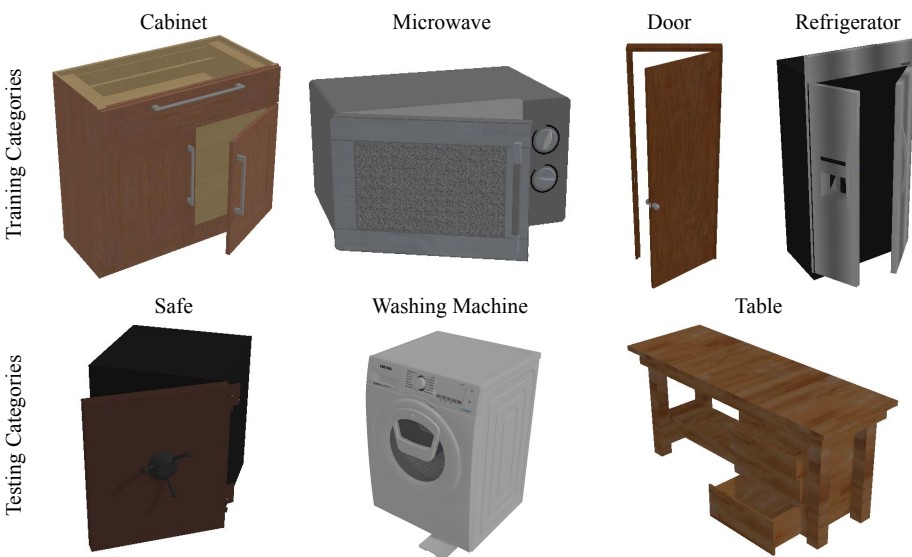

Figure D.6: Data Visualization. We show one example shape from each of the seven object categories we use in our paper.

## D MORE DETAILED DATA STATISTICS

Table D.3 summarizes the data statistics and splits.

Fig. D.6 presents a visualization of some example shapes from our dataset.

## E SAPIEN SIMULATION ENVIRONMENT SETTINGS

We follow exactly the virtual environment settings as in (Mo et al., 2021), except that we randomly sample camera viewpoints only in front of the target shape since most articulated parts are present in the front side of the PartNet-Mobility shapes and it is very unlikely that robot agents need to manipulate object parts from the back of objects. By default, the camera looks at the center of the target shape, and the location is sampled on a unit sphere centered at the origin. For the spherical sampling of the camera location, we set the range of the azimuth angle to [-90°, 90°], and the elevation angle to [30°, 60°]. Please refer to Section A and Section B in the appendix of (Mo et al., 2021) for more in-depth details of the virtual environment.

## F COMPUTATIONAL COSTS AND TIMING

Training of the proposed system took around 3 days per experiment on a single Nvidia V100 GPU. More specifically, the initial RL policy training took around 18 hours, followed by a 15-hour training of the perception module using the RL-collected data. The subsequent curiosity-driven exploration stage took around 10 hours to jointly train the RL and the perception networks. At last, the three prediction branches in the perception module were further fine-tuned for around 20 hours.

During the inference time, a forward pass takes around 15 ms, 8 ms, and 8 ms for the actionability map prediction, diverse trajectory proposals, and success rate predictions, respectively. Our model only consumes 1,063 MB memory of the GPU in test time.

## G   MORE METRIC DEFINITION DETAILS

### G.1   EVALUATING ACTIONABLE VISUAL PRIORS (SEC. 5.1)

We adopt the following evaluation metrics for quantitative evaluation: precision, recall and F-score for the positive data, and the average accuracy equally balancing the negative and positive recalls. Formally, let $f_p$, $f_n$, $t_p$, and $t_n$ denote the data counts for false positives, false negatives, true positives, and true negatives, respectively. The F-score is the harmonic mean of precision and recall, and the other metrics are defined as follow:

$$precision = \frac{t_p}{t_p + f_p},$$

$$recall = \frac{t_p}{t_p + f_n},$$

$$accuracy = \frac{1}{2}(\frac{t_p}{t_p + f_n} + \frac{t_n}{t_n + f_p}).$$

We also compute a coverage score for evaluation, which is calculated as the percentage of ground-truth trajectories that are similar enough to be matched in the predicted trajectories. To this end, we need to measure the distance $D(l_1, l_2)$ between two trajectories $l_1$ and $l_2$, which takes into account both the position difference and the orientation difference at every waypoint. Concretely, we calculate the L1 distance of the waypoint positions as the position distance $D_{pos}$ between the two trajectories. Then, the orientation distance $D_{ori}$ is measured as the 6D-rotation distance of the waypoint orientations. To balance the dimension of the quantities, $D(l_1, l_2)$ is calculated as: $D(l_1, l_2) = 5D_{pos}(l_1, l_2) + D_{ori}(l_1, l_2)$. We consider a ground truth trajectory to be covered if the distance between this ground truth trajectory and a predicted trajectory is lower than a threshold (10, in all our experiments), and then report the percentage of the ground truth trajectories that are covered by the predictions. To compensate the stochastic error, all reported quantitative results are averaged over 10 test runs.

### G.2   EVALUATING DOWNSTREAM MANIPULATION (SEC. 5.2)

For each downstream manipulation task, we locate the contact point sample with the highest action-ability score over the surface on each test shape, and then generate 100 diverse trajectories at this contact point, and pick the trajectory with the highest success rate for executing the downstream task. This results in 350 trajectories for each of the door pushing, door pulling, drawer pushing, and drawer pulling tasks.

In practice, generating trajectories that can perfectly fulfill the given task specification (*e.g.*, push to open a door by 30°) is very hard. Hence, we consider a trajectory can successfully accomplish the task if the specification is fulfilled within a tolerance threshold in percentage $\varepsilon = 15\%$. For example, if the task is to push a door open by 10°, a trajectory that can open the door between 8.5° and 11.5° is counted as a successful sample. We then report the success rate in percentage of all the generated trajectories on each task.

## H   MORE DETAILS OF BASELINES IN SEC. 5.2

In the following, we present more details about the baseline methods that we compare against in Section 5.2.

### H.1   NAIVE RL BASELINE

The naive RL baseline, which is comprised of two sub-RL modules denoted as sub-RL1 and sub-RL2 respectively. Sub-RL1 module takes as input the initial state of the target shape (*i.e.*, the partial scan

of the initial shape, and the mask of the target part), and the task specification, and then outputs the initial position and orientation of the gripper (*i.e.*, the parameters of the first waypoint). Sub-RL2 module takes as input the state of the initial scene (*i.e.*, the partial scan of the shape, the mask of the target part, and the position and orientation of the gripper at the initial state), the state of the current scene (*i.e.*, the partial scan of the shape, the mask of the target part, and the position and orientation of the gripper at the current state), and the task specification, and then predicts the parameters of the next waypoint using the residual representation.

The network architectures of sub-RL1 and sub-RL2 are based on our RL networks as described in Section A.1, with minor adaptations for different input and output dimensionalities. We use a buffer size of 2048 and a batch size of 64 for training both sub-RL1 and sub-RL2 modules. We use the Adam optimizer and a learning rate 0.0001 for the policy and Q-value networks in both modules. We set the initial exploration range to 0.1, which is linearly decayed by a factor of 0.5 every 500 epochs during training.

## H.2    RULE-BASED HEURISTIC BASELINE

For the rule-based heuristic baseline, we hand-craft a set of rules for different tasks. We describe the details as follows.

- **Door pushing:** we randomly sample a point $p$ on the door surface, then initialize the gripper such that the gripper fingertips touch the point $p$ and set the forward orientation along the negative direction of the surface normal at $p$. Let $d$ denotes the distance between $p$ and the rotation shaft of the door, to push the door by a degree $\theta$, we push the door by a degree $\frac{\theta}{4}$ each time and repeat for four times. During each process, we simply move the gripper along the negative direction of the surface normal by a distance of $d\sin\left(\frac{\theta}{4}\right)$.

- **Door pulling:** we initialize the gripper at a point $p$ sampled on the handle of the door and set the forward orientation along the negative direction of the surface normal at $p$. And then we pull the door by a degree $\theta$ by pulling $\frac{\theta}{4}$ each time as in door pushing. Different from door pushing, we also perform a grasping at contact point $p$ for pulling.

- **Drawer pushing:** similarly, we randomly sample a point $p$ on the front board of the drawer, and then initialize the gripper such that the gripper fingertips touch the point $p$ and set the forward orientation along the negative direction of the normal at $p$. Let $t$ denote the push distance in the task specification, we simply move the gripper by a distance of $t$ along the *slide-to-close* direction of the drawer.

- **Drawer pulling:** the middle point of the handle is particularly sampled as the contact point. Note that we count the trial as a failure if the drawer does not have handles. Then we initialize the gripper as in drawer pushing, and move the gripper by a distance of $t$ along the *slide-to-open* direction.

In Fig. H.7, we present some failure cases for the rule-based heuristic baseline to explain why such seemingly intuitive and easy heuristics often fail. See the figure caption for detailed explanations. Such rule-based heuristics require careful human hand-engineering given different task semantics. One needs to hand-design rules for different tasks and sometimes will find it difficult to enumerate all possible rules. Our system, instead, provides a unified system that automatically discovers useful knowledge for different tasks, without the need of spending human efforts, and learn a rich collection of data-driven priors from training over diverse shapes.

## H.3    WHERE2ACT BASELINE

For the Where2Act baseline, we use the pre-trained networks released by Mo et al. (2021) to generate multiple short-term trajectory segments, each of which samples a contact point and an action direction from Where2Act. We assume an oracle part pose tracker and choose the pushing or pulling Where2Act network to query at each time by comparing the ground-truth pose and the goal pose of the target part. Since Where2Act only executes short-term task-less interaction trajectories and does not explicitly consider task degree, we allow it to operate for up to 15 steps and record the closest distance it has ever reached to the desired target part pose. Concretely, in each step, this method

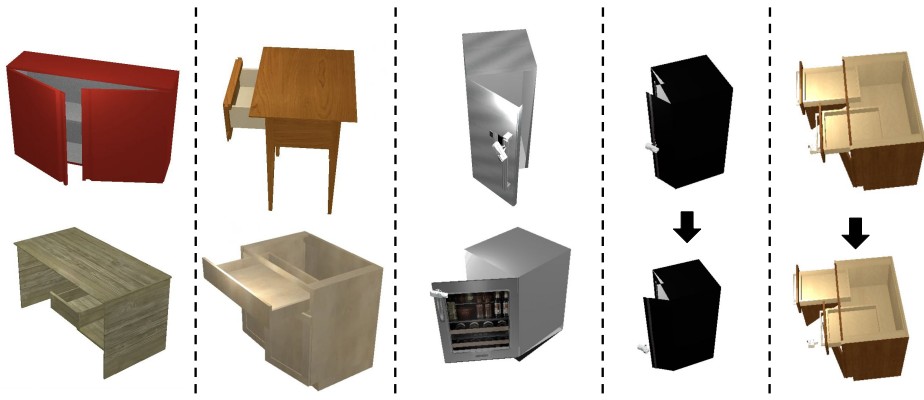

Figure H.7: We present some failure cases for the rule-based heuristic baseline to explain why such seemingly intuitive and easy heuristics often fail. Fist column: these two cabinets have no handle for pulling; Second column: from these viewpoints, these drawer front boards or door handles are occluded; Third column and fourth column: grasps over intricate handle geometry may fail or slip; Fifth column: dynamic factors (*e.g.*, inertia) affects the final achieved part pose, and the drawer moves farther than desired.

Table I.4: We compare our final approach to an ablated version that removes the curiosity-feedback channel and report quantitative evaluations of the visual actionable priors. For each metric, we report numbers over the test shapes from the training categories (before slash) and the shapes from the test categories (after slash). Higher numbers indicate better results.

| | Curiosity | Accuracy (%) | Precision (%) | Recall (%) | F-score (%) | Coverage (%) |
|---|---|---|---|---|---|---|
| pushing door | w/o | 77.70 / 71.73 | 77.83 / 72.62 | 76.99 / 69.89 | 77.55 / 70.80 | **82.35 / 75.52** |
| | w | **82.24 / 72.44** | **81.28 / 72.83** | **85.22 / 73.86** | **82.76 / 72.54** | 82.00 / 70.54 |
| pulling door | w/o | 68.89 / 67.90 | 64.62 / 65.29 | **84.09 / 76.40** | 72.62 / 69.94 | 50.17 / 43.52 |
| | w | **74.01 / 71.31** | **70.52 / 70.26** | **84.09** / 75.85 | **76.06 / 72.01** | **58.68 / 48.29** |
| pushing drawer | w/o | 76.85 / 68.32 | **78.82** / 70.67 | 73.86 / 61.65 | 75.45 / 64.89 | 63.25 / 60.08 |
| | w | **79.69 / 71.59** | 74.65 / **71.80** | **91.19 / 70.45** | **81.65 / 70.52** | **74.15 / 68.08** |
| pulling drawer | w/o | 71.02 / 69.46 | **74.79 / 73.20** | 67.05 / 63.64 | 68.89 / 66.47 | 62.56 / 49.43 |
| | w | **78.41 / 71.88** | 74.54 / 72.29 | **87.50 / 72.44** | **80.23 / 71.71** | **81.15 / 64.31** |

- uses Where2Act's actionability module to predict the affordance of each point, randomly selects a contact point within the estimated top-5 actionable points, and initializes the gripper in front of the selected contact point;

- initializes the gripper's initial 6D pose by running Where2Act's action proposal network and selecting the top-rated action;

- if pushing, directly moves the gripper 0.05 unit-length forward;

- if pulling, moves the gripper slightly forward, closes the gripper to grasp, and moves the gripper 0.05 unit-length backward when the gripper successfully grasps the target part;

- checks if the target part's pose achieves the goal pose, using the target part's ground-truth pose.

# I  ABLATION STUDY ON CURIOSITY FEEDBACK

We compare our final approach to an ablated version that removes the curiosity feedback (Sec. 5.3), in order to validate that the proposed curiosity-feedback mechanism is beneficial.

In Table I.4, we evaluate the prediction quality of the proposed actionable visual priors.

From this table, it is clear to see that the introduced curiosity feedback mechanism (Sec. 5.3) is beneficial as it improves the performance in most entries.

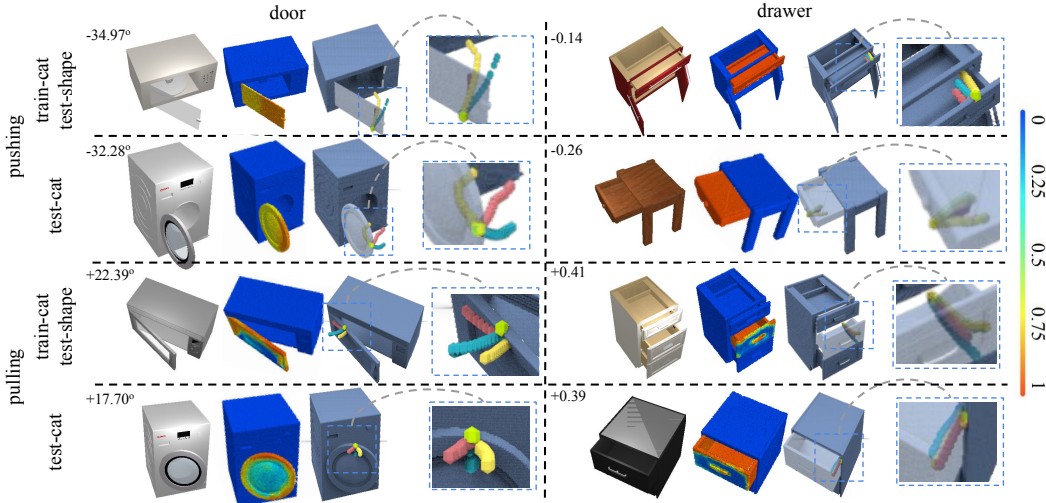

Figure J.8: We show additional qualitative results of the actionability prediction and trajectory proposal modules to augment Fig. 3 in the main paper. In each result block, from left to right, we present the input shape with the task, the predicted actionability heatmap, and three example trajectory proposals at a selected contact point.

## J   MORE RESULTS AND ANALYSIS

In Fig. J.8, we show additional qualitative results of the actionability prediction and trajectory proposal modules to augment Fig. 3 in the main paper.

In Fig. J.9, we present an additional qualitative analysis of the learned trajectory scoring module to augment Fig. 4 in the main paper.

In addition, we provide another set of result analyses in Fig. J.10, where we show that to accomplish the same task, interacting at different contact points over the shape will give different trajectories.

## K   REAL-ROBOT SETTINGS AND EXPERIMENTS

For real-robot experiments, we use one real cabinet and set up one Franka Panda robot facing the front of the cabinet. One ZED MINI RGB-D camera is mounted at the front right of the cabinet. The camera captures 3D point cloud data as inputs to our learned models.

We control the robot using Robot Operating System (ROS) (Quigley et al., 2009). The robot is programmed to execute each waypoint in the predicted trajectory step by step. We use MoveIt! (Chitta et al., 2012) for the motion planning between every adjacent waypoint pair.

We demonstrate various tasks on the cabinet, including pulling open the door at the edge or handle, pushing closed the door at different contact points, pulling open the drawer by grasping the edge or handle. Please check our results in the supplementary video.

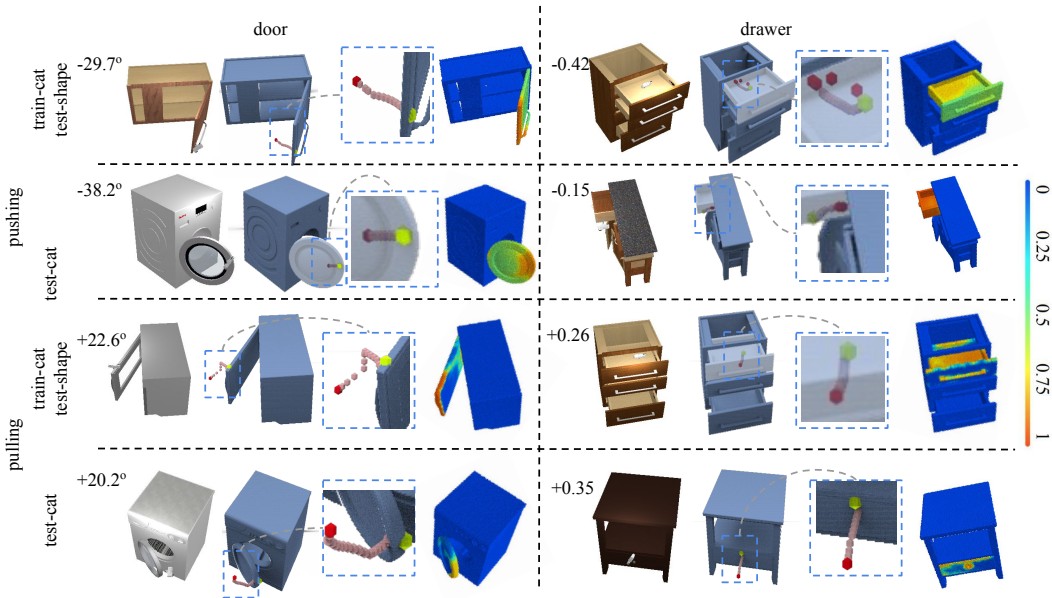

Figure J.9: We present an additional qualitative analysis of the learned trajectory scoring module to augment Fig. 4 in the main paper. In each result block, from left to right, we show the input shape with the task, the input trajectory with its close-up view, and our network predictions of success likelihood applying the trajectory over all the points.

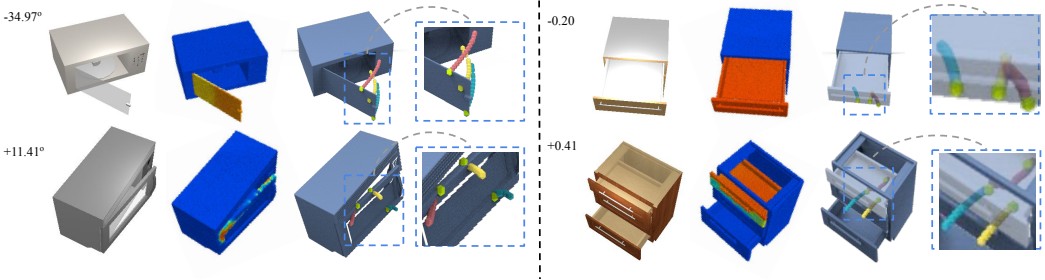

Figure J.10: We use some examples to show that, to accomplish the same task, interacting at different contact points over the shape will give different trajectories. In each result block, from left to right, we show the input shape with the task, the predicted actionability heatmap, and three example trajectory proposals at three different contact points.

