# OpenReview forum: "VAT-Mart: Learning Visual Action Trajectory Proposals for Manipulating 3D ARTiculated Objects"
_ICLR.cc/2022/Conference — ICLR 2022 Poster_

### Official Review · Reviewer_BNFC · 2021-11-02

**Correctness:** 3
**Technical Novelty And Significance:** 3
**Empirical Novelty And Significance:** 3
**Recommendation:** 6
**Confidence:** 3

**Main Review:**

Pros：
The paper proposes the problem of long term action trajectory generation for 3D articulated objects which is not well studied.
The paper is well written.

Cons:
The method itself is not very novel, more about extending the existing Where2Act and a combination of Where2Act and curiosity guidance for RL Policy for Interactive Trajectory Exploration.
Baseline for the Curiosity Guidance for RL Policy for Interactive Trajectory Exploration, which is one of the main components of the method, is not compared in the trajectory generation.


**Summary Of The Paper:**

The paper extends work on static term action generation (Where2Act, ICCV21) for 3D articulated objects to 1) long-term action trajectory generation by learning from data generated data via RL exploration,  2) action trajectory conditioned with task-awareness.

**Summary Of The Review:**

The paper proposes to solve a new problem of long term action trajectory generation for 3D articulated objects, but the method to solve the problem is more about an extension and combination of existing work. The overall quality of the paper, writing and experiment, is good.

---

> ### Author Response · Authors · 2021-11-13
> **REPLY: Thank you so much for the positive feedback!**
>
> Thank you so much for the positive feedback and your constructive suggestions! Thank you for taking the time and effort to review our paper!
>
> We are happy to hear that the reviewer recognizes that 1) the problem our paper proposes is new and worth studying; 2) the overall quality of the paper, writing, and experiments, is good. We sincerely hope that our work can help draw more research attention to this important, challenging, yet under-studied problem.
>
> While we follow the settings of the Where2Act paper, our work attempts to make **an important yet challenging extension to Where2Act** -- learning long-term trajectory predictions to accomplish long-term manipulation tasks. Where2Act only targets very short-term interactions (i.e. the part motion threshold is 0.01) and thus only requires gripper orientation predictions to instantiate short-term hard-coded trajectories. This largely limits its applicability to real downstream tasks requiring manipulating the articulated parts for a mid-term or long-term horizon, such as to open a door by 30 degrees or to fully close a drawer.
>
> There are **several non-trivial challenges to solve on the technical side**, such as how to adapt the Where2Act framework to predict long-term trajectories, how to efficiently collect trajectory data from the simulation, and how to propose diverse trajectories. First of all, trajectory prediction is much harder than predicting simple gripper orientations in Where2Act, due to the much higher degree-of-freedoms of a trajectory than a single 3-DoF gripper orientation. We empirically found that using a waypoint representation, plus leveraging a fixed controller bridging each pair of waypoints, works well in practice. Secondly,  data collection for successful interaction trajectories is much more difficult than Where2Act. While Where2Act simply uses random trials to collect the successful gripper orientations, we observed nearly 0% success rate if we do random trajectory sampling. To address this difficulty, we need to train an RL policy, with carefully designed state space, to collect successful interactions, which is not easy to train across diverse shapes and tasks. Lastly, it is tricky how to make the RL policy generate diverse successful trajectories for supervising our perception networks. We propose to use the perception scores as the feedback rewards, which provides a simple yet effective way to collect diverse trajectory proposals. There are also other small non-trivial changes, such as using cVAE to replace the original Min-of-N strategy proposed in Where2Act as we found that Min-of-N works empirically bad for training trajectory proposals.
>
> For **the “baseline for the Curiosity Guidance for RL Policy”**, if we understand the reviewer correctly, we actually have compared to an ablated version of our framework that removes the curiosity feedback in the appendix (Table I.4). Results in Table I.4 clearly demonstrated that the curiosity feedback mechanism is beneficial as it improves the performance in most entries. Particularly, the curiosity feedback helps the RL policy to generate and the perception system to propose more diverse trajectories in 3 out of the 4 experiments as indicated by the coverage metric. We are eager to hear from the reviewer if we have successfully addressed your concern using this experiment, or we are also happy to add more experiments if requested.
>
> **Thank you again for the useful comments and we hope that we have addressed your questions well. We are happy to take further questions from the reviewer!**

---

### Official Review · Reviewer_1qco · 2021-11-02

**Correctness:** 3
**Technical Novelty And Significance:** 2
**Empirical Novelty And Significance:** 3
**Recommendation:** 6
**Confidence:** 3

**Main Review:**

Overall Comments:
The main novelty of this paper is a slight twist on the Where2Act paper that, instead of generating grip orientations, this paper is generating trajectories.  This difference necessitates the difference in the models.  However, they do show a comparison against the Where2Act approach that shows that their modifications to the network (including the curiosity exploration and how they find the trajectories) are superior, for this task.

This paper is easy to understand and well written.  The ideas are easy to follow and build well.

The appendix was very useful for clarifying some parts of the paper in more detail which was very helpful such as for the curiosity-driven explorations and where the heuristic-based method failed.

Smaller comments:

The term step seems to refer to waypoint step but also can be confused for timestep.  Can you clarify when you use it?

Cite prior cVAE work this is based on in the Trajectory Proposal Module.

**Summary Of The Paper:**

This paper is solving the problem of pushing and pulling objects (mostly things like cabinets) by learning visual action trajectories proposals via a curiosity-driven RL / Perception joint training.  The system input point clouds the object and outputs the actionable score and the per-trajectory success likelihood score on the most likely approach to interacting with the object.  This approach is validated both in simulations but also in simulation and with real results.

**Summary Of The Review:**

This paper has some interesting elements to it and their approach is validated by both real and simulated results.  While the task isn’t very novel, they at least validated their approach to show that it does better than previous approaches thus contributing to the field.  They also do a good job of explaining each of the steps in the appendix to make it easy to understand what exactly they are doing.  For these reasons, it merits inclusion in the conference.

---

> ### Author Response · Authors · 2021-11-13
> **REPLY: Thank you so much for the positive feedback!**
>
> Thank you so much for the positive feedback and the suggestions for further improving our work! Thank you for taking the time and effort to review our paper!
>
> It is our great pleasure to hear that the reviewer recognizes our extensions beyond Where2Act, our new architecture designs, and our superior performance than Where2Act. We are very thankful to know that the appendix provides useful clarifications and experiments in addition to the main paper. We sincerely hope that our work can inspire future works on this important, challenging, yet under-studied problem.
>
> Our work attempts to make an important yet challenging extension to the Where2Act paper -- learning long-term trajectory predictions to accomplish long-term manipulation tasks. Where2Act only targets very short-term interactions (i.e. the part motion threshold is 0.01) and thus only requires gripper orientation predictions to instantiate short-term hard-coded trajectories. This largely limits its applicability to real downstream tasks requiring manipulating the articulated parts for a mid-term or long-term horizon, such as to open a door by 30 degrees or to fully close a drawer. It brings non-trivial challenges to solve on the technical side, such as how to adapt the Where2Act framework to predict long-term trajectories (which is much harder to predict than a simple gripper orientation), how to efficiently collect trajectory data from simulation (where we need to train an RL policy to collect successful interactions, which is more difficult than using random trials in Where2Act), and how to propose diverse trajectories (for which it requires the curiosity-driven RL rewards to generate diverse successful trajectories).
>
> We are also very happy to hear from the reviewer that the paper is easy to read and the ideas are presented well. **Below we address your smaller comments and have revised the paper to reflect the changes**. Thank you for raising up these points to make the paper clearer.
>
> > The term step seems to refer to waypoint step but also can be confused for timestep. Can you clarify when you use it?
>
> Thank you for pointing this out and this is a great point! Yes, in our paper, we always use the term “step” to refer to the “waypoint step”, not the “timestep”, as we do the predictions at the level of waypoints and rely on a fixed controller to bridge the execution steps. We have revised the paper to make this clearer.
>
> > Cite prior cVAE work this is based on in the Trajectory Proposal Module.
>
> We mostly follow the plain cVAE [Paper-1] by designing a pair of conditional trajectory encoder and decoder in the Trajectory Proposal Module. Our trajectories are just sequences of waypoints, so it is fairly simple to just concatenate the five waypoints at maximum into a vector. While previous works have also demonstrated using cVAEs to predict future pedestrian [Paper-2] or car trajectories [Paper-3], our case is much easier so we directly employ a plain cVAE. We have revised the paper to include the citations.
>
> **Thank you again for the useful comments and we are happy to take further questions from the reviewer!**
>
> - [Paper-1]: Sohn, Kihyuk, Xinchen Yan, and Honglak Lee. “Learning Structured Output Representation using Deep Conditional Generative Models.” Advances in Neural Information Processing Systems. 2015.
>
> - [Paper-2]: Yao, Yu, Ella Atkins, Matthew Johnson-Roberson, Ram Vasudevan, and Xiaoxiao Du. "Bitrap: Bi-directional pedestrian trajectory prediction with multi-modal goal estimation." IEEE Robotics and Automation Letters 6, no. 2 (2021): 1463-1470.
>
> - [Paper-3]: Feng, Xidong, Zhepeng Cen, Jianming Hu, and Yi Zhang. "Vehicle trajectory prediction using intention-based conditional variational autoencoder." In 2019 IEEE Intelligent Transportation Systems Conference (ITSC), pp. 3514-3519. IEEE, 2019.

---

### Official Review · Reviewer_dxTr · 2021-11-05

**Correctness:** 4
**Technical Novelty And Significance:** 3
**Empirical Novelty And Significance:** 2
**Recommendation:** 6
**Confidence:** 4

**Main Review:**

I have two main concerns regarding the paper:

1)The link between the visual perception and the RL policy appears weak as the only feedback is through exploration rewards for the RL policy to try out interactions on places where the visual perception models assigns low success probability. But is this a good exploration bonus? What if indeed these are simply not good places to act and succeed? Shouldn’t the certainty of the visual model be taken into account as opposed to low probability?

2)The naïve RL baseline seems to be training one RL policy across multiple tasks (if I understood correctly) that operates on a point cloud input. This baseline is designed to fail. It can be much improved by training a separate RL policy in each environment separately, and simply disabling the visual perception representations? Then, at the end, we can train actors that operate directly from images, similar to an ``asymmetric actor critic" setup.





**Summary Of The Paper:**

The paper proposes a method for exploration of 3D articulated environments that alternates between collecting interaction data with RL while maximizing a combination of extrinsic and intrinsic rewards, and training visually conditioned action maps, image conditioned manipulation trajectory priors, and success predictors, that further guide the intrinsic reward prediction during data collection.

**Summary Of The Review:**

Could the authors show results of the baseline suggested above?
Could the authors explain the rational of the exploration reward?

It is possible that I am missing something from my understanding of the paper. I will be careful during discussion period to clarify any misunderstandings.



Post rebuttal : the authors have put together the requested baseline and they show significant performance margins over it. Thank you very much for this, I raise my score accordingly.

---

> ### Author Response · Authors · 2021-11-13
> **REPLY: Thank you! We have clarified some misunderstandings, addressed your concerns, and hope to hear back from you if you have further questions! [part 1/2]**
>
> Multipart response [part 1/2]
>
> Thank you for taking the time and effort to review our paper! And, thank you for your questions.
>
> **First of all, it seems to us that there are very likely some important misunderstandings regarding the problem setting of our work and we would like to clarify. Our paper does NOT aim for exploring the 3D articulated environments, but we focus on training a perception system that predicts visual actionable information over 3D shapes. The RL policy is only intended to be used during training for collecting successful interaction trajectories to supervise the perception networks. We only use the perception system at the test time to predict the intended visual actionable information over 3D shape as the outputs of the system.** We are sorry for the misunderstandings and would like to incorporate changes in the writing to make things clearer if requested.
>
> Our proposed VAT-Mart framework takes a 3D partial scan of an articulated object as input and outputs **point-level visual actionable information over the 3D shape**, including an affordance map, per-point trajectory proposals, and per-trajectory success ratings, for different manipulation tasks (e.g., pushing closed doors, pulling open drawers). **The visual actionable information predicted by our perception system can serve as informative action priors guiding potential downstream robotic planners and controllers.** For example, given a novel unseen cabinet with a closed door and an arbitrary target degree to open the door, one can directly obtain the visual actionable priors by feeding the input 3D scan of the shape through the learned perception system. The predicted visual actionable priors will highlight possible locations and propose likely trajectories for accomplishing the task even before you do any planning or interaction. The downstream robotic planner, controller, or RL manipulation policy can benefit from such visual priors for more efficient exploration or learning.
>
> **Below, we address the two main concerns you raised. Please do not hesitate to reply to us if you need further clarifications and discussions. We are eager to hear back from you!**
>
> > The link between the RL policy and the visual perception networks are weak. Only linked by the exploration feedback? Why train RL over places with low success ratings? They may be bad places to act? Why not use visual model certainties as rewards?
>
> We first want to make it clear again that the RL policy is only used during training to collect successful interaction trajectories to supervise the perception networks. **the RL policy is primarily trained with the task extrinsic rewards** (described in the “Reward Design” paragraph of Sec. 4.1) **to accomplish the target task** (e.g., pull open a cabinet door by 30 degrees, push closed a door for 0.1-unit-length), since the only purpose for the RL policy is to search for trajectories that accomplish a given manipulation task over a given 3D shape and pass the data to the perception sides as supervisory signals. The RL policy does NOT take the action maps predicted by the perception networks as inputs to guide the trajectory searching, since we want to explore possible interaction trajectories at all locations over the articulated parts of the 3D shape.
>
> We think that **there may be a confusion between the action maps suggesting what _places_ to act and the success ratings for the interaction _trajectories_**. We use the latter as the exploration reward, not the former. We are not encouraging the RL to try at the unlikely actionable _places_ (as the reviewer says, we shouldn’t). Instead, we use the negatives of the _trajectory_ success ratings as the RL intrinsic rewards to encourage the RL to propose more novel trajectories that the perception networks have not seen yet. More intuitively, for each RL sampled interaction trajectory successfully accomplishing a given task, we check if the perception system has already assigned the trajectory with a high score. If so, it indicates that the perception system has already learned this trajectory so we do not need to feed too much of this data again to the perception networks during training. If a low score is predicted for the trajectory, it means that it is a new knowledge to the perception systems and thus we want to encourage the RL policy to propose more trajectories like this to train the perception system better.

---

> ### Author Response · Authors · 2021-11-13
> **REPLY: Thank you! We have clarified some misunderstandings, addressed your concerns, and hope to hear back from you if you have further questions! [part 2/2]**
>
> Multipart response [part 2/2]
>
> > Why not train the RL baseline in each environment separately and simply disable the visual perception representations? Using the “asymmetric actor critic” setup?
>
> We first want to make sure that we understand your question correctly. When talking about “multiple tasks/environments”, do you mean “different shapes”? There could be other similar things confusing the terms “tasks/environments” such as different manipulation tasks (e.g. push closed a door, pull open a drawer) and manipulation target tasks (e.g. push closed the door by 10, 20, or 30 degrees). We are sorry about this and would like to clarify more clearly in the paper revision if needed.
>
> If you mean “different shapes”, we provide our response below. Otherwise, please do not hesitate to reach out and we would like to address your question in other cases.
>
> We cannot train separate RL policies in different environments since we aim to obtain an RL policy that can operate across different shapes and can be directly tested given a novel unseen shape at the test time. Firstly, this is to make sure that the comparisons to our method are fair since our VAT-Mart framework, once trained, can be directly tested over novel unseen test shapes without further training. Secondly, we do not want to learn a single RL policy from scratch for any test-time shape we encounter, which is also very likely impossible in the real-world cases since 1) we do not have a fully simulatable 3D shape mesh of the shape (i.e. the complete shape mesh with annotated parts and joint information) to load into the simulator and train the system in simulation, and 2) performing real-world interactions using real robots to train an RL from scratch using real-world trial-and-errors is too expensive.
>
> For the “asymmetric actor critic”, are you referring to this paper “Pinto, Lerrel, Marcin Andrychowicz, Peter Welinder, Wojciech Zaremba, and Pieter Abbeel, _Asymmetric actor critic for image-based robot learning_. Robotics: Science and Systems 2018”? To our best understanding of this work, it seems that they are not training an RL over different shapes/geometry? Our work uses the setting of training a single system that works across different shapes/geometry and aims for directly generalizing to novel shapes at the test time. Also, if we understand correctly, the complete states for training the critic come from the simulator in that paper? Then, for novel real-world 3D shapes at the test time, we may not have access to the full 3D meshes with articulated parts and joints annotated for the simulation.
>
> **We hope our replies above help clarify our paper more clearly to you. Please do not hesitate to reply to us if you have further questions! We are eager to hear back from you! Thank you!**

---

> ### Author Response · Authors · 2021-11-30
> **Thank you**
>
> Thanks for raising your rating to 6. We are glad that our responses help alleviate your concerns. Thanks again for all your valuable feedback!

---

### Decision · Program_Chairs · 2022-01-20

**Decision:**

Accept (Poster)

**Comment:**

The paper claims to present actionable visual representations for manipulating 3D articulated objects. Specifically, the approach learns to estimate the spatial affordance map as well as the trajectories and their scores. After checking the rebuttal from the authors, all reviewers agree that the paper adds value to the research area. In the end, it got three borderline accept ratings. The initial criticism included lacking (experimental) comparison to baselines, and the authors successfully corresponded to the request from the reviewer. One reviewer commented that the proposed approach is a combination of Where2Act and curiosity guidance for RL Policy for Interactive Trajectory Exploration, which we believe is a valid point. Still, the paper extends the previous Where2Act and successfully demonstrates its success on difficult tasks.

We recommend accepting the paper.